# Pre-Service Teachers' Experience of Learning about Sustainability in Technology Education in South Africa

Asheena Singh-Pillay 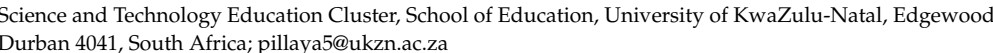

Science and Technology Education Cluster, School of Education, University of KwaZulu-Natal, Edgewood, Durban 4041, South Africa; pillaya5@ukzn.ac.za

**Abstract:** There have been many calls for the integration of Education for Sustainable development in the training of pre-service teachers so that they can develop sustainability action competence to address sustainability issues now and in the future in their communities. This qualitative pilot project sought to explore pre-service technology teachers' (PSTTs) experiences of learning about sustainability in the processing of the polymers module, when they were engaged in community-based assessment tasks. Data was collated from 25 PSTTs enrolled for the processing of the polymers module using reflective diaries and semi-structured interviews. Thematic analysis revealed that PSTTs had three key experiences of learning about sustainability, making a difference, learning as hands-on, minds-on, and hearts-on and sustainable pedagogies. At a theoretical level, the findings highlight the value of adopting an ESD lens and experiential learning approach to infuse learning about sustainability when teaching technology education. Further, the findings indicate that if PSTTs are challenged to participate actively in sustainability issues in their communities, they make informed choices about their role in society as future teachers, the pedagogies they plan to adopt, and the kinds of learning they strive to promote in learners.

**Keywords:** education for sustainable development; contextualized learning; experiential learning; sustainability; sustainability competence

## 1. Introduction

Globally there have been many calls for the integration of Education for Sustainable Development (ESD) in the training of pre-service teachers, for example, Decade of Education for Sustainable Development (2004–2014) [1], UNESCO's Global Action Programme (2015–2019): Sustainability begins with teachers [2] and Agenda 2030 [3] with its 17 sustainable development goals. Within the South African context, similar mantras for the integration of ESD are articulated by the White Paper on education and training [4] and the National Curriculum Statement [5]. These calls see education as a conduit for change, not just for ESD but also for the learning process and developing sustainable thinking among learners. They have placed teacher educators in a liminal space to transform communities and initiate change via the curriculum and their teaching. To take on the role of drivers of ESD and to propagate sustainable thinking in the training of pre-service teachers, teacher educators need to rethink and reimagine the curriculum, embrace pedagogies that allow for the integration of ESD in modules, reflect on practice, plan assessments which allow pre-service teachers to engage in sustainable thinking without being prescriptive. Sinakou et al. [6] assert that beyond pre-service teachers being exposed to content during teaching, they need opportunities to participate in sustainable thinking and sustainability action. Active student participation in the learning process can be seen as quintessential for sustainable learning and thinking. It entails reflecting and deconstructing existing ways of knowing and doing to help the student to adapt to changing circumstances and critically assess problems they confront. Such transitions in the curriculum ought to provide the space for pre-service teachers to develop sustainability competencies, an ethic of care towards

environmental issues, and agency to solve sustainability issues in their local community and consequently empower them to embrace ESD in their teaching to increase the uptake of ESD at a school and community level [7,8].

I am a teaching educator at a South African University where this research was conducted. I lecture to pre-service teachers specializing in teaching technology education (henceforth referred to as PSTTs). The module I have lectured to PSTTs for the last 3 years is the processing of polymers (the third content module that PSTTs must complete). The module deals with the chemistry involved in the processing of polymers (plastics). I had "inherited" this module from a colleague when I was appointed at the university. The processing of the polymers module was structured and developed prior to my arrival at the university.

The module content and assessment activities were lecture-based. Traditional lecturing approaches were used, which did not challenge students to participate actively, make decisions, and reflect on sustainability issues. Such approaches did not provide opportunities for PSTTs to embark on real sustainability action [5], nor did they foreground the development of skills and values PSTTs needed to address environmental and sustainability issues (the monomers of plastics and plastic products are crude oil), despite the immense problem South Africa faces with the reckless disposal of plastic bags. South Africa generates 2. 4 million tons of plastic waste annually [9]. For example, the task linked to the properties of the different categories of plastic was a quiz. The content and the assessment activities were not designed or structured for PSTTs' learning to be linked to action in the environment to resolve sustainability issues faced by the local communities and to possibly initiate in PSTTs citizenship responsibility, care, or agency towards the environment and sustainability issues.

The theoretical nature of the assessments and the lack of opportunity for experiential learning in the processing of the polymers module warranted the following questions to be raised: how should the processing of the polymers module be revamped to be environmentally and socially responsive to South Africa's huge problems of poor waste disposal, wasteful consumption, urban decay, pollution, the decline in biodiversity, climate change and increasing poverty? And how could PSTTs be empowered to promote sustainability in a socially responsible manner in their classrooms and communities without being prescriptive?

The above questions prompted reflection on the existing knowledge-driven approach to the module activities; consequently, an ESD lens was adopted to enable the activities to be contextualized. Contextualized learning links learning to the learner's community context and their prior learning [10]. It allows learners to deconstruct existing knowledge to create their own solutions to real sustainability issues [11]. It facilitates deep learning and reflection on the learners' day-to-day experiences, environment, and community [12]. I decided to initiate a pilot project in the processing polymers module, which entailed a revamping of the assessment activities within the module. The assessment activities were linked to the community and involved learning from real sustainability actions and experiences. The research findings of Leal Filho's [13] study emphasize the need for academics to make decisions in their lecture rooms to propagate sustainable thinking and sustainability. For example, given South Africa's problem of poor disposal of plastics, in the revamped task on the properties of the different categories of plastics, PSTTs had to engage in a community-based research project in one of the five communities identified by them that were near the university. For the community-based project, PSTTs were required to work in self-selected groups (5 PSTTs per group) in their selected community: Firstly, PSTTs were required to conduct an audit of plastic used in 5 randomly selected households in their community. For the audit, PSTTs had to establish the number of plastics used per day, week, and month, note the type of plastic used, how the plastic was disposed of, note landfills in the community, illegal dumping, pollution of river/streams, recycling-upcycling, and burning of refuse in the community. The rationale for the audit was to focus on the consumption of plastics by the communities, as everyone consumes something on a daily basis [14]. Paying attention to consumption allows learners the chance

to unpack and reflect upon the values and beliefs that motivate these individuals, as well as others', habits and behaviors pertaining to consumption. Secondly, PSTTs had to share their audit finding with the community. Thirdly PSTTs had to work with the community to jointly address the sustainability challenges they had identified. Fourthly PSTTs (in their group) had to present their research project findings (with detailed evidence) at a student poster presentation arranged within the technology discipline. Based on the revamped community-based assessment task on the properties of the different categories of plastic, this research thus responded to the following question: What are PSTTs' experiences of learning about sustainability during the community-based task in the processing of the polymers module?

Gaining insight into PSTTs experiences in learning about sustainability is important as PSTTs will eventually qualify and become the drivers of the UNESCOs Global Action Plan (2014a), as well as Agenda 2030 [3]. Mawela [15] emphasized that future teachers must be able to reach out to their community by contextualizing the curriculum to address local issues of sustainability. Furthermore, Durrani et al. [16] noted there is a paucity of practice-led research in technology education on the integration of ESD in pre-service teacher training curricular and pre-service teachers' experience of learning about sustainability in technology education. The research intends to respond to the gap identified in the literature.

This paper is organized into five sections. The introduction is contained in the first section. The second section pays attention to the literature review, the third to the methodology, the fourth to findings and discussion, and the fifth section is the conclusion.

## 2. Literature Review

This section focuses on integrating education for sustainable development in higher education institutions and sustainability competence.

### 2.1. Integration of Education for Sustainable Development at Higher Education Institutions

The call for the integration of Education for sustainable development in higher education institutions has been supported by [1] UNESCO's Decade of Education for Sustainable Development (2005–2014), the Global Action Programme on Education for Sustainable Development (post-2014) [2] and Agenda 2030 with its 17 sustainable development goals [3]. However, these calls are dependent on the individual academics, values, and motivation to integrate and spread learning about sustainable development among their students [17]. Integrating issues on sustainable development require a change in the module content [18], pedagogy [19], and forging links with the community so students can undertake assessments relating to sustainability in their communities [20]. Scholars such as [21,22] assert it is important to align theory and assessments with real contextual issues pertaining to sustainability so that students can develop social responsibility and critical citizenship to address issues of sustainability locally and globally [23]. This means that in the absence of such motivation and values in teacher educators, it would not be possible to integrate ESD into their module content and adopt pedagogies that facilitate learning about sustainability and developing values among students. Thus, the goal of higher education institutions to support students to respond to sustainability issues and to initiate critical thinking about their beliefs and attitudes towards sustainability locally and globally would be unattainable.

### 2.2. Sustainability Action Competence

It is quintessential for teacher educators to arrange the content and their pedagogy in a manner that facilitates the enhancement of students' knowledge, skills, and competence needed to address sustainability issues [24]. Put simply, teaching educators need to use teaching approaches that link assessments to the community. This is so that students encounter an action-oriented experience in their communities [25] to work with real problems related to sustainability and thereby develop sustainable thinking and action competence for sustainability [24,26]. The action-orientated experience involves acting on

real sustainability issues, which requires students to get actively involved and make well-informed decisions about their learning and interaction with peers and the community [6]. Such action-orientated activities and interaction in the community allow for a change in students' thinking, feeling, attitude and behavior when dealing with environmental and sustainability issues in their community [27].

*2.3. Experiential Learning*

Kolb's [28] Experiential Learning Theory (ELT) framed this study. This study focused on PSTTs' experiences of learning about sustainability in the processing of the polymers module, and the ELT model was an apt framework for the following reasons: Firstly, it illuminates the shift in experiences which results in learning and provides an explanation of the learning that underscores the journey to sustainability. Secondly, PSTTs are involved in a learning process that involves all senses, emotions, and values, thinking, reflecting, and acting [21]. Thirdly, experiential learning allows for opportunities for transformative experiences that result in transformative learning for sustainability when the PSTTs experienced all four stages of ELT. Fourthly, ELT is effective for teaching students about sustainability in consumption [29].

The ELT model, as shown in Figure 1 below, has two opposing stages of understanding experience, namely concrete experience (CE) and abstract conceptualization (AC), and two opposing stages of transforming experiences, namely reflective observation (RO) and active experimentation (AE). When PSTTs engaged with their community-based assessment task, they underwent the four stages of the ELT (1) having a concrete experience that resulted in (2) observation of and reflection on that experience, which led to (3) the integration of the reflections, to make sense of the experience and to conceptualize possibilities to intervene in the situation and to (4) apply the new ideas during active experimentation. In other words, via the four stages of experiential learning, the shift made during the levels of learning is visible, for example, from doing things (CE-first level learning) to doing this better (RO-second level learning) to seeing things differently (AC-third level learning–epistemic learning).

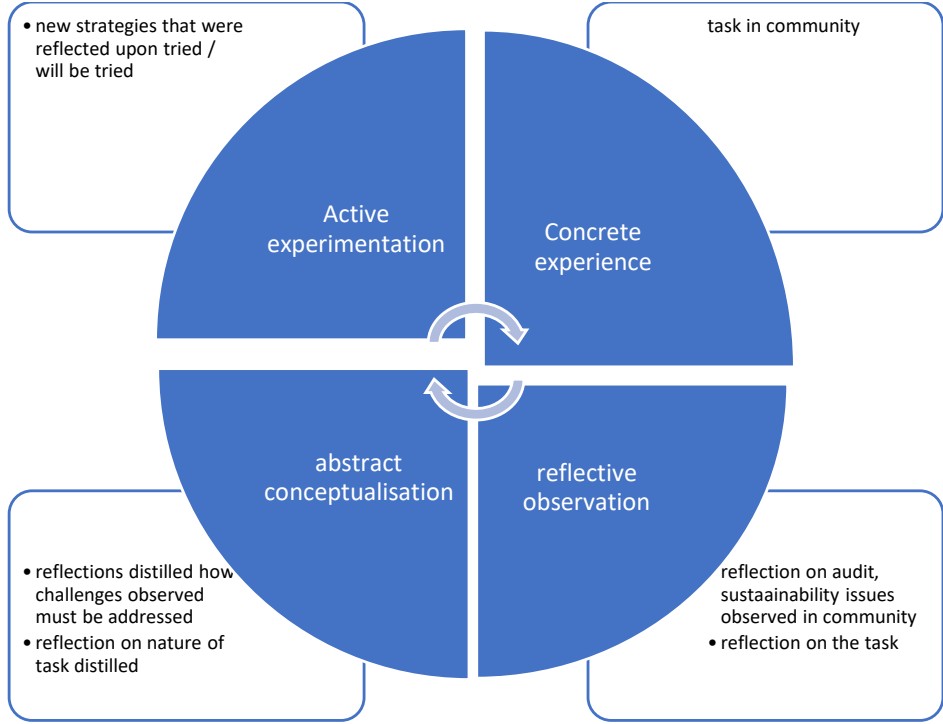

**Figure 1.** Kolb's experiential learning cycle– source authors.

Participatory pedagogies that promote critical self-reflection that lead to transformed habits of the mind are the essence of transformative learning [30].

## 3. Materials and Method

### 3.1. General Background

Guided by the interpretative paradigm, this study used a qualitative approach. The qualitative approach focused on non-numeric data and was thus best suited to obtain insights into the experiences of PSTTs in learning about sustainability in the processing of the polymers module [31]. Data was collated at one South African University. PSTTs who enrolled for the processing of the polymer modules in 2018 consented to take part in the study. Permission to conduct the study was obtained from the university ethics office and relevant gatekeepers.

### 3.2. Participants

The 25 PSTTs enrolled for the processing of the polymers module in 2018 were invited to participate in this study. An informed consent letter, which outlined the research aims, design, ethical protocol to be adhered to, and the voluntary nature of their participation, was given to the participants. All 25 PSTTs agreed to participate in the study (15 females and 10 males). PSTTs self-selected five groups to work in the five identified communities around the university. This means 5 PSTTs worked in each of the five identified communities. The communities were given the pseudonym 1–5. Prior to the start of their research project, the 25 PSTTs were coached on how to embark on participatory action research (PAR). PAR was used as a data generation method. It allowed PSTTs to gain confidence in addressing social issues in their community and examine their beliefs about teaching and as teacher-researchers.

### 3.3. Instruments

Prior to data collection, PSTTs chose a number from 1 to 25, which was recorded by the PSTT and the lecturer to note which PSTTs belonged to groups A to E. Data was collated via a reflective diary and individual semi-structured interviews. The number (1–25) selected by the PSTT as their pseudonym was allocated to their reflective diary and interview transcript. PSTTs were capacitated on how to keep a reflective diary. The brief was to reflect on their observations (regarding the audit of plastic used per household per day, week, and month in their community, the type of plastic used, how the plastic was disposed of, landfills illegal dumping, pollution of river/streams, recycling-upcycling, and burning of refuse in the community), emotions, thinking when they interacted with their community and their experience of learning about sustainability in the processing of the polymers module. Each PSTT maintained a reflective diary.

The interview with the PSTTs focused on two issues: namely, PSTTs' experience of learning about sustainability and their sustainability actions, as well as their experience of working with their communities to address sustainability issues. For this research, only PSTTs experiences of learning about sustainability and their sustainability action will be analyzed and considered. The individual interview was 20 min in duration, and each interview was audio recorded. Each group must collaboratively produce a poster that displays the sustainable issues confronted by the community and action taken to resolve the issue (with evidence) and present their research project findings at a student poster presentation arranged within the technology discipline.

### 3.4. Data Analysis

The interviews were transcribed and sent to the respective interviewee to check if their responses were captured accurately, which is known as respondent validation and is a way of ensuring the credibility of the data. Data from the reflective diaries and interviews were exposed to thematic analysis. The interview transcripts and the reflective diaries were read several times to note patterns, convergences, and divergences before coding could start [31].

The codes assigned initially were reviewed, and data and initial codes from the interviews as shown in Table 1, and reflective diaries were juxtaposed and thereafter re-grouped to refine the emergent themes. The emergent themes were also returned to PSTTs to check for accuracy and resonance with their experiences.

**Table 1.** Showing categories and codes for an interview question.

| Interview Question | Category | Codes |
|---|---|---|
| Experience in Learning about sustainability using PAR in communities | Making a difference: Gatekeeper of sustainability | Difference, transform, unique, change, appreciated, unlike other, empower |
| | Learning: Hands-on | Activity apply, action, physical, real |
| | Minds on | Problem-solving, critical thinking, reflection, troubleshooting, challenged thinking |
| | Hearts on | Emotion, care, empathy, deep concern, fore relations |
| | Sustainable pedagogies | Contextualized, ESD lens, transformative, action research |

## 4. Findings and Discussion

In this section, findings from the PSTTs audits conducted at communities 1–5, followed by their experiences of learning about sustainability in the processing of the polymers module, are presented.

### 4.1. Community Audit

The audit conducted in the five communities (1–5) by the PSTTs is reflected in Table 2 below. An average of 20 plastic shopping bags per week were used by the households sampled in the five communities. The most frequently used plastic item is the plastic shopping bag. The most common way, in all five communities, to dispose of plastic used is by burning and burying it. The communities seem unaware of the harmful effect that burning plastic has on the environment and the health of people. Nor do they seem aware that plastics are non-biodegradable, so burying them will not result in their decomposition. Regarding recycling, a common practice in the five communities is to reuse plastic containers (high-density polyethylene) for the storage of food and household items. Residents in community 4 are involved in upcycling plastic shopping bags to make baskets, mats, and bins, which they sell at craft and flea markets to generate income. Strong, sturdy plastic material is used by the residents of community 4 (mainly informal dwellings) to waterproof the roof and walls of their homes. Only two communities have landfill sites. The rivers/streams in all five communities are filthy and polluted.

**Table 2.** Findings of audit in 5 identified communities.

| Community | 1 | 2 | 3 | 4 | 5 |
|---|---|---|---|---|---|
| Average use of plastics per household per week | 15 | 25 | 10 | 30 | 20 |
| Most common type of plastic used | Plastic shopping bag (low density polyethylene) | Plastic shopping bags | Plastic shopping bags | Plastic shopping bags | Plastic shopping bags |
| Disposal of plastics | Burning burying | Burning, burying, dumping anywhere | Burning burying | Burning burying dumping | Burning, burying, dumping anywhere |
| Recycling | Reuse of shopping bags, margarine tubs, ice cream tubs, for storage of grains, sugar, tea, leftover food | Reused for storing things | Reused for storing things | Weaving of shopping bags to make mats, bins, baskets for sale at flea markets | Strong plastic used for water proofing dwellings |
| Number of landfills in community | 0 | 0 | 1 | 0 | 1 |
| Pollution of rivers/streams | Polluted with shopping bags, take away containers, household garbage excrement, rubble | Detergents as clothes are washed at rivers/streams, garbage, plastics, building rubble | Plastic household garbage | Plastics, car oil, clothes | Plastics, dead animal bodies |

Table 2 above highlights that the most common type of plastic used in these five communities is the plastic shopping bag. This means that for the members of these five communities buying plastic bags when they shop is a part of their lives. Their use of plastic shopping bags has not decreased, despite having to pay for these bags, which are meant for single use. This particular find resonates with that of O'Brien and Thondhlana's [32] study, which reported the extensive use of plastic shopping bags by South Africans even though the levying taxes has increased.

These five communities' efforts to dispose of plastics by burning, burying, illegal dumping, and pollution of rivers/streams illustrate their poor management of plastic disposal together with a lack of awareness of the hazards of these methods used to dispose of plastics to the environment and their health and a lack of awareness of the effects of microplastics on the environment. The above finding echoes the findings of [33], which emphasized that in South African communities' plastics are a problem because of the lack of education about the disposal of plastic, poor waste management, and the dangers plastic poses to the environment and human health.

The audit conducted by PSTTs undergirds their experiences of learning about sustainability in the processing of the polymers module. These experiences are presented and discussed next.

### 4.2. PSTTs Experience of Learning about Sustainability in the Processing of the Polymers Module

Analysis of the data from the reflective diaries and the interview show that PSTTs had three core experiences of learning about sustainability in the processing of the polymers module, viz;

- Making a difference: gatekeepers of sustainability
- Learning as minds-on, hands-on and hearts-on
- Sustainable pedagogies

Each of these core experiences is discussed next.

### 4.3. Making a Difference: Gatekeepers of Sustainability

Responses from the reflective journals and interviews indicate that all 25 PSTTs found they made a difference in their communities. The difference PSTTs made arose from their

actions, the assistance, and knowledge they were able to provide to the community to improve their level of awareness of the consequences of burning, burying, and illegal dumping of plastics, the need for upcycling of plastic used and the dire need to reduce their excessive use of plastic shopping bags. PSTTs felt they served as gatekeepers of ESD in their communities and had a responsibility to perpetuate pro-sustainable behavior, wise consumption, sustainable actions, and attitudes among their young and old community members:

> *It was a unique experience as I realized that teaching extend beyond the classroom. I felt appreciated by the community, my actions made a difference when they asked me to suggest how they could reinforce and waterproof their informal dwelling, reduce the amount of plastic they use and how to dispose waste (PSTT interview 9).*

> *I'm realizing we have to spread ESD, this research project was different, and unlike other assessments I have encountered, I want to construct similar actionable tasks in my classrooms so learners could be empowered to be aware of sustainability problems and care for the local environment and planet. (PSTT interview 3).*

> *Children are our future, they have to be taught and empowered how to solve environment problems, learning to care about the environment and how to address sustainability issues does not happen by accident, children have to engage in tasks which allow them to solve sustainability issues in their surroundings, they must be involved in active action learning . . . this is what I take from this empowering module, I can make a difference via my teaching . . . (PSTT interview 17).*

Likewise, the reflections, from the reflective diaries, mirrored similar testimonies:

> *I see myself as part of my community, I did not see myself in this way before this activity, I have to get them to break free from their excessive use of plastic shopping bags, I have to gatekeep, monitor, and transform unsustainable activities and actions (PSTT reflective diary 11).*

> *It is my duty and responsibility to care for my community and the environment and teach and empower them about sustainability issues (PSTT reflective diary 23).*

The above testimonies confirm PSTTs' experience of making a difference in their community. In making a difference in their communities, PSTTs realized that teaching is not confined to the classroom but extends to the community. Further, PSTTs accept that learning and acting on sustainability issues requires more than just knowledge to portray sustainability behavior. The excerpts above reveal PSTTs acknowledge the need for community-based tasks to develop sustainability competence, such as critical thinking, and change in their mindset and reflections. PSTTs in this study accept that teachers have a moral and social dimension and a responsibility to transform unsustainable consumption and actions in their communities. Similar findings were noted by Ates and Gül [34] study, which highlighted that students needed more than knowledge and skills to display sustainable behavior and quintessential skills such as critical thinking and reflection. Thus, when PSTTs created awareness, knowledge, and skills in their communities on their use of plastics and the need for the community to care for the environment, they demonstrated the action competence needed for sustainability. Action competence for sustainability is the ability to act in terms of one's own capacities to contribute to change and be influential [35].

PSTTs' experience of engaging in the community-based task has initiated in them willingly to embrace their critical citizen responsibilities, which allowed them to forge ties with the community and effect change. In the process of making a difference in their community, they served as gatekeepers of sustainability (reducing the amount of plastic they use and how to dispose of waste). These above findings confirm that actively engaging PSTTs with community-based tasks allowed for critical reflection. The leverage that the community-based task has is illuminated as it enlightens PSTTs about the social and moral dimensions of teaching and their responsibility to address social issues. This change in awareness about their role and responsibilities is important as it helps them to make a difference in their outlook on sustainability. The above findings resonate with those of Laurie,

Tarumi, McKeown, and Hopkins' [36] study, which emphasizes that pre-service teachers who are exposed to ESD in their program embrace a culture of sustainability, develop sustainable competence and care for the environment among the different communities in which they teach. Tasks that embrace active engagement in the community can be used to foster in PSTTs sustainability action competencies.

*4.4. Learning as Minds-On, Hands-On, and Hearts-On*

All twenty-five PSTTs reported their experience of learning about sustainability in the processing of the polymers module was hands-on, minds-on, and hearts-on.

*The task extended learning to the community, we had a chance to actively apply the content learnt in class to solve real contextual problems, learning was hands-on (PSTTs interview 4).*

*We had a chance to think critically, problem-solve, and troubleshoot in our communities This module, and the task challenged our thinking and actions about sustainability in our communities (PSTTs interview 1).*

*The processing of polymers module opened my mind, eyes and heart, I gave me a chance to reflect on my role as a teacher and my responsibilities to civic issues, helping to uplift my community (PSTT interview 16).*

Reflections from the reflective diary corroborated the testimonies from the interviews:

*This module in particular the activity in the community has changed my thinking-about the type of tasks to set to engage learners, my role as a teacher, promoting sustainability, my actions–getting involved to solve local problems, using my role as a teacher to bring change (PSTT reflective diary 25).*

*That the task gave me the space to learn to care, show compassion for the community, environment, gain confidence to teach about sustainability, forge relations with the community and classmates, learning about sustainability issues is now close to my heart (PSTT reflective diary 18).*

*I now think critically, solve problems linked to the everyday life of the community, collaborate with their classmate to resolve the sustainability issues identified in our community. Realized that sustainability is everyone's responsibility (PSTT reflective diary11).*

The above excerpts highlight that PSTTs' learning began when they were actively involved with the processing of polymers community-based task. PSTTs acknowledge that the previous style of assessment that they were exposed to is not applicable to learning about sustainability.

Engagement with the community-based task in the polymer module provided PSTTs with concrete experience in hands-on learning.

While the task was hands-on, at the same time, the task was "minds-on" as it afforded PSTTs the space to think critically, work collaboratively and reflect deeply on their role as teachers, the kinds of the task they plan to set for their learners, their responsibility as teachers to the community and the environment. Furthermore, PSTTs were able to think critically, participate in abstract conceptualization and apply content knowledge learned in lectures to actively experiment and problem-solve sustainability issues in the community. PSTTs found the task to be meaningful as they could link theory to real-life situations in the community. Moreover, PSTTs reported changes in their mindsets, frames of reference, confidence levels, and attitudes toward the community and environment. Thus, the learning PSTTs experienced was experiential and transformative (hearts-on). The above findings indicate that learning about sustainability is not just a cognitive (minds-on) process. It embraces a practical aspect (contextualized task, hands-on) and an affective component that allows for transformation (hearts-on). Sinakou et al. [6] assert that action-oriented learning opportunities allow learners to develop their competencies as responsible citizens

and make well-informed decisions on real sustainability issues in society. The findings of this study are in line with that of Mahmud's [37] study conducted in Malaysia, which asserts that learning about sustainability facilitates personal experience for participants resulting in profound changes in knowledge, skills, and attitudes. The ESD approach used in the processing of the polymer modules has resulted in positive changes in PSTTs' attitudes to their citizenship responsibilities.

### 4.5. Sustainable Pedagogies

Twenty-three PSTTs reflected in many ways that they have learned sustainable pedagogical approaches to teaching technology education:

> *I had never been exposed to contextualized learning before this module, I have learnt to think differently about pedagogy after this task on the processing of polymers, I have come to learn the community is a great resource for engaging learners in problem-based learning, project-based leaning, action research, I will use contextual learning to teach technology education. (PSTTs reflective diary 24).*

> *I was not exposed to ESD, I think it is very important that everyone learns about it, I will integrate ESD into all the topics in the technology curriculum for example in structure I will integrate sustainable consumption of natural resources when building thatch roofs,-learners have to be taught about ESD, they have to experience and see how issues of sustainability impact their life-so I will use an ESD lens as pedagogy in my teaching to bring about change in learners knowledge, skills, attitudes, values (PSTTs interview 11).*

> *I will try participatory action research, project-based learning, in communities, also transformative learning –it worked well in this module and I enjoyed learning, working with the community, I have come to realize that teachers are drivers of sustainability through their pedagogy. The pedagogy used can bring out change in learners' attitudes, care for the environment (PSTTs reflective diary 19).*

PSTTs drew on their experience and exposure to the processing of polymers community-based task as a basis to establish the pedagogy they will embrace in their classroom practice. Via the processing of the polymers module, PSTTs realized that to learn about sustainability issues, learning needed to be grounded in the sustainability issues confronted by the community daily [38] PSTTs realized that it is crucial to use interactive strategies such as contextualized learning, project-based learning and action research in a community setting to promote meaningful student learning, and chances for reflection to bring about change in values, skill and behavior as well as care for the community and environment. The pedagogies PSTTs espouse to embrace include exposing learners to components of critical thinking, decision-making, reflection, and value-based learning, all of which are action competence for sustainability. Such pedagogies will train learners to act currently and in the future on sustainability issues [39]. An important point to note is that PSTTs realized that learning is not limited to the classroom, nor is it the transfer of content from teacher to learner. Rather, it is collaborative, situated, and a social process [40].

### 5. Conclusions

This research explored PSTTs' experiences of learning about sustainability in the processing of the polymers module. Analysis of data from reflective diaries and the semi-structured interviews indicate that PSTTs had three core experiences of learning about sustainability in the processing of the polymers module. Firstly, PSTTs learned that they could make a difference as gatekeepers of sustainability. This finding shows that if pre-service teachers are equipped with content knowledge and are exposed to experiential learning that foregrounds sustainability, they can make a difference and contribute to reform in sustainability. The above findings concur with that of Ates and Gül's [34] study, which emphasizes students require more than content knowledge to display sustainable behavior, and vital sustainability action competencies can only arise through action-oriented

tasks [6]. The sustainability action competencies such as critical thinking, problem-solving, troubleshooting, reflection, making informed decisions, care, compassion, and recognition of their citizenship and social responsibilities that PSTTs gained via experiential learning allow them to serve as gatekeepers of sustainable thinking and sustainability. PSTTs acknowledge their social responsibility as teachers to address sustainability issues. In adopting their role as gatekeepers of sustainability, they will be shaping their community, society, and the world in a sustainable fashion.

Secondly, PSTTs experienced learning as hands-on, minds-on, and hearts-on. The community-based task allowed PSTTs to apply knowledge and skills learned in lectures *to* a real-world context and provided them with the platform to see the relevance of the content studies in class with sustainability issues identified in the community. The community-based task ensured continuity of learning from two fronts. Firstly, PSTTs acquired an understanding of sustainability issues affecting their community, such as waste disposal, use of plastic bags, pollution, and recycling. Secondly, PSTTs developed values, attitudes, and social responsiveness, which contributed to their personal growth. The above finding unveils the kind of learning PSTTs appreciate, namely, active learning strategies that are hands-on, minds-on, and hearts-on. Arising from the above finding, a recommendation is made that novel reimagined pre-service technology education programs are a fundamental part of capacitating PSTTs for sustainability. Regarding the above recommendation, Tuncer et al. [41] noted that teachers will only produce students who are sustainability literate if they themselves are knowledgeable and have sustainability action competencies.

In terms of sustainable pedagogies, it is clear that PSTTs plan to conduct their future teaching using pedagogies that allow students to gain knowledge, skills, and sustainable competencies, by engaging in action-oriented experiences in community settings which allow for sustainable thinking and for learners to come up with solutions to problems. PSTTs envisage that this kind of engagement will allow learners to gain sustainability competencies to cope with sustainability issues in the present and in the future. The sustainable pedagogies that PSTTs espouse are experiential and transformative. The findings of this study clarify that action-oriented experiences are key to initiating in PSTTs sustainable action competencies, such as critical thinking, problem-solving, reflection, lifelong learning, and decision-making for the present and future. These ideas are supported by [6], who posit that action competence is a prerequisite for the development of sustainable competencies.

At a theoretical level, the finding of this pilot study indicates that adopting an ESD lens and experiential learning approach to teach the processing of the polymers module allowed PSTTs to connect theory with practice when they were engaged with a community-based task. It allowed them to develop sustainable thinking skills, the habit of mind, disposition to become a more active and rational learner. Traversing through the four stages of experiential learning allowed PSTTs to make decisions about their learning, interact with peers and the community, critically reflect on and evaluate sustainability issues in the community, as well as their role as critical citizens and drivers of ESD. Thus, a recommendation is made for community-based activities to be integrated into technology modules to allow for community engagement and the development of academic and sustainability action competencies in PSTTs. This means that teaching should be aimed at supporting and encouraging the learner to adopt the approach to learning which requires active participatory engagement and reflecting on one's learning and thinking.

**Funding:** This research received no external funding.

**Institutional Review Board Statement:** Not applicable.

**Informed Consent Statement:** Informed consent was obtained from all subjects involved in the study.

**Data Availability Statement:** All data are available from the corresponding author upon reasonable request.

**Conflicts of Interest:** The author declares no conflict of interest.

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
