# Peer review of "Pre-Service Teachers’ Experience of Learning about Sustainability in Technology Education in South Africa"

_sustainability, doi:10.3390/su15032149_

Round 1

Reviewer 1 Report

I appreciate the integration of an action-focused/experiential element into an existing course. Engaging preservice teachers in the kinds of pedagogies and practices  needed to promote sustainability literacy in K-12 contexts is essential, but challenging given the traditional structure of most higher education courses.

Wendy Wakefield is an emerging US-based scholar whose commitments to enhancing sustainability education for preservice teachers is similar to your own.

I'd appreciate more explanation of the audits.  How were the "randomly selected households" identified and selected? What was the time frame for the audit - how many days, weeks, etc? How was the refuse obtained and from whom? 

I do not think Table 1 is necessary. 

I enjoyed reading and reviewing this manuscript. Thank you for the opportunity.

Author Response

Please find the attached response. Thank you for your suggestions.

Reviewer 2 Report

The pre-service teachers' conceptions of STE studied in the present research are relevant. The qualitative methodology used in the current study is appropriate for this type of research, and literature review and data analysis are applicable. However, in conclusion, it would be relevant to specify the limits of the results put forward at the methodological and conceptual levels.

Author Response

(The authors gave the same response as above.)
